# Trajectories and Forces in Four-Electrode Chambers Operated in Object-Shift, Dielectrophoresis and Field-Cage Modes—Considerations from the System’s Point of View

**DOI:** 10.3390/mi14112042

**Published:** 2023-10-31

**Authors:** Jan Gimsa, Michal M. Radai

**Affiliations:** 1Department of Biophysics, University of Rostock, Gertrudenstr. 11A, 18057 Rostock, Germany; 2Independent Researcher, HaPrachim 19, Ra’anana 4339963, Israel; michal.radai@gmail.com

**Keywords:** AC electro-kinetics, electro-kinetic object manipulation, inhomogeneous object polarization, microchambers, micro-systems, object manipulation, field-cage, μTAS, MatLab^®^ model, thermodynamics, energy dissipation, LMEP

## Abstract

In two previous papers, we calculated the dielectrophoresis (DEP) force and corresponding trajectories of high- and low-conductance 200-µm 2D spheres in a square 1 × 1-mm chamber with plane-versus-pointed, plane-versus-plane and pointed-versus-pointed electrode configurations by applying the law of maximum entropy production (LMEP) to the system. Here, we complete these considerations for configurations with four-pointed electrodes centered on the chamber edges. The four electrodes were operated in either object-shift mode (two adjacent electrodes opposite the other two adjacent electrodes), DEP mode (one electrode versus the other three electrodes), or field-cage mode (two electrodes on opposite edges versus the two electrodes on the other two opposite edges). As in previous work, we have assumed DC properties for the object and the external media for simplicity. Nevertheless, every possible polarization ratio of the two media can be modeled this way. The trajectories of the spherical centers and the corresponding DEP forces were calculated from the gradients of the system’s total energy dissipation, described by numerically-derived conductance fields. In each of the three drive modes, very high attractive and repulsive forces were found in front of pointed electrodes for the high and low-conductance spheres, respectively. The conductance fields predict bifurcation points, watersheds, and trajectories with multiple endpoints. The high and low-conductance spheres usually follow similar trajectories, albeit with reversed orientations. In DEP drive mode, the four-point electrode chamber provides a similar area for DEP measurements as the classical plane-versus-pointed electrode chamber.

## 1. Introduction

This work is the third in a series of papers on the dielectrophoresis (DEP) behavior of high and low conductivity 2D spheres in square chambers driven by various combinations of idealized pointed and plane electrodes. In previous work, the chamber was energized with pointed-versus-plane electrodes [1] and plane-versus-plane and pointed-versus-pointed electrodes [2]. We have calculated the DEP force and the corresponding trajectories from the system’s point of view via an approach that uses the positional dependence of the total dissipation [3]. This approach is consistent with the law of maximum entropy production (LMEP) [4,5,6,7]. It assumes that the dissipation in the system increases at a maximum rate along each DEP trajectory.

One advantage of the approach is that it accurately accounts for the inhomogeneous field distributions in the chamber and in the object due to the influence of such structures as chamber walls and electrodes on the field distribution in the object, as well as the influence of the object’s presence on the field distribution in the chamber, e.g., due to mirror charges. Another advantage is that the approach to calculating the DEP force uses the relatively simple numerical calculation of “the system’s total conductance” (hereafter referred to as “the system’s conductance”) from the distributions of currents and potentials. The force acting on the object at a given location is then calculated from the gradient of the change in the conductance of the system without the need for integration over the volume or surface area of the object. For details, see discussions in: [1,2,8,9].

Here we complete our considerations using configurations with four-pointed electrodes placed in the center of the square chamber’s edges. The four electrodes were operated in three drive modes; object-shift mode (two adjacent electrodes opposite the other two adjacent electrodes), DEP mode (one electrode versus the other three electrodes), and field-cage mode (two electrodes on opposite edges versus the two electrodes on the other two opposite edges).

## 2. Theory

From the system’s point of view, the work done on a volume of material can be stored as electric or magnetic field energy [3]. Our model uses the third possible mechanism, the dissipation of electrical energy according to the Rayleigh dissipation function [10], which is proportional to the conductance of the DEP system. According to the LMEP, the direction of the fastest increase in dissipation, which in our system corresponds to the fastest increase in the conductance of the system, determines the direction of thermodynamic evolution of the system, and thus the direction of DEP translation [2,5].

It is generally accepted that DEP forces arise from the interaction of the object’s induced in-phase (real) polarization components with the inducing external field. Accordingly, from the system’s point of view, the DEP force is calculated from the in-phase contributions to the energy dissipation. Unfortunately, in the overall system, in-phase contributions to energy dissipation can also arise from the interaction of the out-of-phase components of the currents with out-of-phase components of the induced polarizations. Moreover, these contributions to the total dissipation may depend on the object’s position. DEP force calculations from energy differences must exclude such contributions, which are not trivial [9]. We use DC properties for the external and object media to avoid the occurrence of out-of-phase components in the polarizations and currents. It should be mentioned that another way to avoid out-of-phase components simply is to consider the capacitive charge work for the whole system for the high-frequency limit [9]. By combining appropriate DC conductivities for the external medium and the object, the model allows analysis of any real polarization relationship of the external medium and the object that may occur at a given frequency for frequency-dependent properties.

The method used below to calculate forces and trajectories has been discussed in previous works [1,2,9]. In short, 2D “conductance fields” were calculated for each electrode and sphere/medium conductance configuration. The conductance fields describe the dependence of the conductance of the system on the position of the sphere’s center. Thus, the trajectory of the maximum conductance gradient can be found for each starting position of the sphere. Depending on the structure of the conductance field, DEP trajectories with different start positions lead to the same or different endpoints.

The gradient in the system’s conductance for position ${\vec{r}}_{i}$ was derived numerically from the vector ${\vec{r}}_{i+1}-{\vec{r}}_{i}$, which was normalized to the step width $∆r=\left|{\vec{r}}_{i+1}-{\vec{r}}_{i}\right|$, and points in the direction of the maximum increase in the sheet conductance $L_{DEP}^{2D}$:(1)$${grad}_{NUM}\left(L_{DEP}^{2D}\right)=MAX\left(\frac{{∆L}_{DEP}^{2D}}{∆r}\right)\frac{{\vec{r}}_{i+1}-{\vec{r}}_{i}}{∆r}$$

To compare the DEP forces in different chamber setups, a relative DEP force was derived by normalizing the force to the square of the chamber voltage, the depth of z=1 m perpendicular to the sheet plane, and $L_{Basic}^{2D}$, the sheet conductance of the chamber without object:(2)$${\vec{F}}_{DEP}^{2D}\sim \frac{Z}{L_{Basic}^{2D}}MAX\left(\frac{{∆L}_{DEP}^{2D}}{∆r}\right)\frac{{\vec{r}}_{i+1}-{\vec{r}}_{i}}{∆r}$$

This is Equation (8) from [2].

## 3. Materials and Methods

We consider 200-µm 2D spheres suspended in a 1 × 1-mm square chamber, as in previous work. The 2D conductances of the spheres and the external medium differ by a factor of ten, combining 1.0 S with 0.1 S and vice versa. The 1 × 1-mm chamber area consisted of 199 × 199 2D voxels, with only the central 160 × 160 voxels accessible to the center of the sphere. More details on the geometries of the chambers and objects can be found in the previous publications [1,2,9].

For each electrode array and each pair of sphere and medium conductance, the conductance of the chamber was calculated for each of the voxels accessible to the center of the sphere using a MatLab^®^ routine, resulting in “total conductance matrices” (hereafter referred to as “conductance matrices”) with 160 × 160 elements [11]. The conductance matrices were used to derive “conductance fields”, using the matrix values as interpolation points for the MatLab^®^ quiver line function. The conductance fields are available in Appendix A. More details about the software used for the calculations can be found in the previous publications. For better visibility of the inhomogeneous polarization of the sphere in the respective figures, equipotential line and current line plots were calculated with a MatLab^®^ routine. Within the conductance fields, the sphere’s center follows trajectories along the conductance gradient, i.e., each step increases the conductance of the DEP system and hence the dissipation of electric field energy.

Figure 1 shows the empty chambers’ equipotential and current line distributions with three different driving modes. In the following, the driving modes in A, B, and C are denoted as (++−−), (+−+−) and (+++−), respectively, corresponding to the potentials at the four electrodes. While the field in mode B has only one mirror symmetry line and no rotational symmetry, modes A and C have more than one mirror symmetry line and one or more rotational symmetry angles.

## 4. Results and Discussion

### 4.1. General

Table 1 shows the calculated chamber conductances for the three drive modes of the chambers without and with the 2D sphere. The table summarizes the effect of the presence of the spheres at different positions on the periphery of the chamber, at intermediate positions and at the center position. In each drive mode, the conductances of the chambers are increased or decreased in the presence of the 2D spheres with high or low-conductance, respectively, compared to the empty chambers. In each case, the magnitude of the effect, and thus the electrical work done in the chamber, depends on the sphere’s position.

The basic, maximum and mean conductance values were taken or calculated from the 199 × 199 conductance matrices (Appendix A), while the conductance values for positions A to H were obtained from a MatLab^®^ routine for calculating the field distributions.

Misalignment between the extrapolated µm positions and the underlying 199 × 199 2D voxel grid are the leading cause of any slight numerical discrepancies between the minimum and maximum conductance values and the conductance values for the corresponding object positions in Table 1; for example, compare the minima and maxima in (+++−) mode with the conductance values at positions A and C. In the following, the effect of the sphere on the current and equipotential line distributions is considered in detail.

### 4.2. (++−−)—Drive Mode (Object-Shift Mode)

#### 4.2.1. Field Distribution and Chamber Conductance

Figure 2 shows results for the 1.0-S sphere in 0.1-S medium. Figure 2A–C show sphere positions at the edge of the chamber; Figure 2D–F are intermediate positions with respect to the center position in Figure 2G. Note that the positions in Figure 2A,D have three siblings, i.e., three different positions with the same (mirror or rotational) symmetric field distributions. The positions in Figure 2B,C,E,F each have a sibling with mirror symmetric field distributions.

Figure 2A–F reflect the most general case of inhomogeneous polarization of the sphere in an inhomogeneous external field, as can be seen from the current and equipotential lines. In Figure 2G, the presence of the sphere distorts the external field, which is mainly homogeneous without the conducting sphere (see Figure 1A). The current flows mainly in the upper left and lower right caps in the sphere. The equidistant current lines at x = 0 are the reason for the seemingly asymmetric current line distribution e.g., in the upper right corner of Figure 2C. This effect did not occur with equidistant current lines on the diagonal from the upper left to the lower right corner of the chamber.

The conductance of the chamber increases in the order w/o < C < F < B < G < E < D < A, where w/o is the conductance without the sphere (Table 1). A and C are the most and least favorable of the seven positions, according to LMEP. The seven conductances of A-G are elements of the 160 × 160 conductance matrix (Appendix A), which were used as interpolation points to generate the corresponding conductance field.

Figure 3 shows results for the 0.1-S sphere in a 1.0-S medium. For the low-conductance sphere, the same chamber positions are considered as in Figure 2. The symmetry and sibling properties are the same as in Figure 2. Again, the polarization of the sphere is inhomogeneous in Figure 3A–F. However, in Figure 3G, it appears to be largely homogeneous. As in Figure 2, the equidistant current lines at x = 0 are the reason for the slight asymmetries in the current line distributions in the external medium.

The conductance of the chamber increases in the order A < E < D < G < B < F < C < w/o, where w/o is the conductance without the sphere (Table 1). According to the LMEP, C and A are the most and least favorable of the seven positions. The seven conductances of A–G are elements of the 160 × 160 conductance matrix (Appendix A).

#### 4.2.2. Trajectories and Forces

Figure 4 and Figure 5 show conductance fields with trajectories for the high and low-conductance spheres, respectively. The 19-voxel wide, white frames in Figure 4A and Figure 5A are geometrically inaccessible to the sphere’s center. In Figure 4B,C and Figure 5B,C, sheet conductance and normalized DEP force are plotted over the same abscissas. In both conductance scenarios, the chamber conductance increases monotonously along each trajectory toward specific endpoints (Figure 4B and Figure 5B). The normalized DEP forces in Figure 4C and Figure 5C were calculated using Equation (2).

The diagonals of the chamber from low left to top right and from top left to low right are a mirror plane for conductances, trajectories, and forces, respectively (Figure 4A). Both diagonals are watersheds that divide the chamber into four triangular regions of attraction for the four stable endpoints E_1_, E_2_, E_3,_ and E_4,_ near the electrodes. These endpoints are located slightly away from the electrode tips because the sphere experiences a lateral bias generated mainly by the attraction of the nearest counter electrode. This is evident from the current lines in Figure 2A, where the sphere at the left electrode is displaced toward the upper counter electrode. Three unstable endpoints (E_5_, E_6_ and E_7_) are saddle points located on the inverted mirror plane. The trajectories within the mirror planes and the triangular planes have one sibling (a_1_, c_1_, c_2_ and d_1_) and three siblings (b, e, f, g, h, i, and j), respectively. The trajectories a_2_, c_3_ and d_2_ are three of four siblings.

The three saddle points on the inverted mirror plane create a complex conductance field structure. As a result, in most cases, the sphere does not move along the shortest possible trajectory to the endpoints. From the corners, the sphere is deflected toward the center of the chamber before the trajectories are bent toward one of the nearby electrodes, e.g., trajectories e, f, g, h, i, and j. Trajectories starting between like-charged electrodes, e.g., e, j, and g, have a larger arc than trajectories starting between counter electrodes, and may even pass the endpoint and move back to the endpoint along the chamber wall.

For improved clarity, trajectory c has been divided into c_1_, c_2_, and c_3_. Section c_1_ starts precisely in the corner and ends at the unstable saddle point E_5_. A slight disturbance may cause it to run along the vertical watershed to one of the unstable saddle points E_6_ or E_7_ (here c_2_ toward E_6_), where the sphere can be deflected almost perpendicularly to either side (c_3_ or a_2_) and hit the chamber wall near the electrode, creating a small force peak (Figure 4C). A higher force peak is generated before the sphere reaches the end point. The terminal steps lead to negligible changes in the chamber’s conductance, resulting in minimal forces, according to Equation (2). As noted in previous work [1], the highest force peaks are generated when the electrode is hit directly, which is almost the case with trajectory e. The DEP force is zero at the unstable saddle points E_5_, E_6,_ and E_7_, but not at the stable endpoints E_1_, E_2_, E_3,_ and E_4_, where the sphere’s motion stops. Experimentally, the DEP force at the stable endpoints is compensated for by the wall pressure.

The conductance field in Figure 5A has similar symmetry properties to that in Figure 4A, but with the force directions and trajectories reversed. Force peaks are observed when the sphere detaches from the chamber wall. The forces are higher the closer the starting point is to an electrode. Again, trajectories in the volume have three siblings each, e.g., a, c, e, f, and g. Trajectories on the two diagonals have only one sibling, e.g., b_2_, b_3_, d, h_1_ and h_2_. However, the four stable endpoints are located distant from the electrodes in the corners of the chamber, and there is no diagonal watershed from low left to top right. On the other diagonal, a watershed runs only between the two unstable saddle points E_6_ and E_7_, with the third unstable saddle point E_5,_ halfway corresponding to the center of the chamber. The bifurcations at both ends of the diagonal watershed make E_6_ and E_7_ triple points, each with three catchment areas, i.e., an instability at E_6_ (E_7_) can deflect the object to one of the three endpoints E_1_, E_2_, or E_4_ (E_2_, E_3_, or E_4_). In either case, it is theoretically possible for the sphere to move along the watershed and stop at E_5_ or to pass E_5_ and continue to E_2_ or E_4_. In the chamber plane, the three watersheds form two pairs of mirror symmetric catchment areas, one pair of mutually distant small triangular-like areas with the endpoints E_1_ and E_3_ and another neighboring pair of large pentagon-like areas with the endpoints E_2_ and E_4_.

Comparison of Figure 4 and Figure 5 shows that the conductance fields for the high and low-conductance spheres have similar symmetry structures. For example, a line representing a watershed in one of the two figures is the central trajectory of a bundle of trajectories in the other figure. One example is the trajectories h_2_ and b_3_ in Figure 5, which correspond to the mirror plane in Figure 4. Another example is the trajectories d and h_1_ in Figure 5, which correspond to the trajectories a_1_ and d_1_, respectively, in Figure 4. In Figure 5, the trajectories h_1_ and d start at the unstable saddle point E_7_ in opposite directions, h_1_ along the watershed and with d as the central trajectory of a bundle of trajectories with the end point E_3_ (cf. trajectories c and f with end point E_1_).

Trajectories between the like-charged electrodes in the larger catchment areas of the endpoints E_2_ and E_4_, e.g., a, e and g, have larger arcs than trajectories between counter electrodes bounded by the curved watersheds. Along the watersheds, the trajectory b is divided into sections b_1_, b_2_, and b_3_. Section b_1_ originates at the top edge of the chamber and runs along the curved watershed. For numerical reasons, b_1_ deviates slightly from the watershed before E_6_, so it turns left at E_6_ and continues as b_2_ to E_5_, where it turns left again toward E_2_. The trajectory h is divided into sections h_1_ and h_2_, which are siblings of b_2_ and b_3_, respectively. Section h_1_ starts from the saddle point E_7_ and turns left at E_5_ toward E_4_.

In both conductance scenarios, a 90° rotation of the electrode drive voltages reverses the orientation of the trajectories in the center region of the chamber (Figure 4 and Figure 5). Thus, the corresponding switching of the drive voltages could be used to manipulate the position of the object in the chamber.

### 4.3. (+++−)—Drive Mode (DEP Mode)

#### 4.3.1. Field Distribution and Chamber Conductance

In “DEP mode”, one electrode is driven against the other three electrodes. Figure 6 shows the results for the 1.0-S sphere in a 0.1-S medium. The sphere positions were chosen similarly to the (++−−)—drive mode. However, the DEP (+++−)—drive mode has only one horizontal mirror symmetry line, which reduces the number of sibling positions. While positions C, D, G and H are unique, positions A, B, E and F have one sibling.

Without the sphere, there is hardly any location in the chamber with a homogeneous external field (Figure 1B), and the sphere is clearly inhomogeneously polarized at all positions shown in Figure 6A–G. However, comparing the changes in the local fields along the central axis of the chamber from C to D, we can expect that the object must be nearly homogeneously polarized at position H, where the object’s presence creates largely symmetric local and external fields.

In Figure 6, the conductance of the chamber increases in the order w/o < A < H < E < G < F < D < B < C, where w/o is the conductance without the sphere (Table 1). C and A are the most and least favorable of the seven positions, according to LMEP. The conductances of the eight positions A–G are elements of the 160 × 160 conductance matrix (Appendix A).

Figure 7 shows the results for the 0.1-S sphere in a 1.0-S medium. For the low-conductance sphere, the same chamber positions are considered as in Figure 6. The symmetry and sibling properties are the same as in Figure 6. Again, the polarization of the sphere is inhomogeneous at all positions.

With the sphere in the upper right corner (not shown), the conductance is 325.2 mS, i.e., position A is a better hiding position where the presence of the sphere has a smaller effect on the conductance. Clearly, the sphere’s presence at position A has the least influence on the conductance of the chamber and the difference between A and w/o is minimal. The conductance of the chamber increases in the order C < B < D < F < G < H < E < A < w/o (Table 1). A and C are the most and least favorable of the seven positions, according to LMEP.

The conductances of the eight positions are elements of the 160 × 160 conductance matrix (Appendix A).

#### 4.3.2. Trajectories and Forces

Figure 8 and Figure 9 show conductance fields with trajectories for the high and low-conductance spheres, respectively. In Figure 8B,C and Figure 9B,C, sheet conductance and normalized DEP force are plotted over the same abscissas. In both conductance scenarios, the chamber conductance increases monotonically along each trajectory toward specific endpoints (Figure 8B and Figure 9B). The normalized DEP forces in Figure 8C and Figure 9C were calculated using Equation (2). The horizontal centerline is a mirror plane for watersheds, conductances, trajectories, and forces (Figure 8A and Figure 9A).

For the high-conductance sphere, three watersheds divide the chamber into four catchment areas with stable endpoints on the electrodes (E_1_ and E_3_) or near the electrodes (E_2_ and E_4_) (Figure 8A). The latter two are located somewhat away from the electrode tips because the sphere experiences a lateral bias generated mainly by the attraction of the single negative electrode. Each watershed has one saddle point (E_5_, E_6_, and E_7_). E_5_ is an unstable saddle point located at the intersection of the mirror plane and the curved vertical watershed.

Trajectories a and b within the mirror planes have no siblings; all other trajectories have one sibling. Trajectories a and b start from the unstable saddle point E_5_ on the symmetry line into opposite directions, straight to E_3_ and E_1_, respectively. Except for these trajectories, the sphere does not move along the shortest possible trajectory to the endpoints. From the corner regions, the sphere is deflected to the center of the chamber before the trajectories are redirected toward one of the nearby electrodes, e.g., trajectories c, d, f, i, and k. Trajectories starting in the catchment areas of E_2_ and E_4_ can pass the endpoint and then move along the chamber wall back to the endpoint, e.g., f and i. E_3_ at the single counter electrode has the largest catchment area, and the trajectories ending at E_3_ can be long, e.g., e, j, and l.

The highest force peak is generated when trajectory a hits the electrode directly. However, the force peak of trajectory b is generated in an area where the conductance changes are very moderate, and the forces generated are small (Figure 8B). Small additional force peaks are generated when the sphere hits the chamber wall near the electrode before moving along the wall to the endpoint (Figure 8C, trajectories c, j and h). Another higher force peak is generated before the sphere reaches the endpoint. The terminal steps within the conductance field often lead to negligible changes in the chamber’s conductance, resulting in very small forces. The DEP force is zero at the unstable saddle points E_5_ (Figure 6H), E_6_, and E_7_, but not at the stable endpoints E_1_, E_2_, E_3_, and E_4_, where the motion of the sphere comes to a stop, and the DEP force is (experimentally) compensated by the wall pressure.

The conductance field for the low-conductance sphere in Figure 9A has similar symmetry properties to that in Figure 8A but with reversed force directions and trajectories. Accordingly, force peaks are observed when the sphere detaches from the chamber wall. The forces are higher the closer the starting point of a trajectory is to an electrode (Figure 9C). The DEP force is zero at the saddle points E_5_, E_6_, and E_7_ but not at the stable endpoints E_1_, E_2_, E_3_, and E_4_.

A straight watershed on the mirror symmetry line and two symmetrical, curved watersheds divide the chamber into two pairs of catchment areas with the stable endpoints E_1_, E_2_, E_3_ and E_4_ at the corners of the chamber. While the straight watershed is the borderline between the two large catchment areas with stable endpoints E_1_ and E_4_, the two curved watersheds bound two catchment areas with stable endpoints E_2_ and E_3_. Each watershed has one saddle point (E_5_, E_6_, and E_7_).

The unstable saddle points E_6_ and E_7_ are in slightly different locations than for the high-conductance sphere in Figure 8. E_5_ is at the same location for both conductance scenarios. The trajectories a and b along the mirror symmetry line have no sibling, and all other trajectories have one sibling. From areas near an electrode, the sphere moves within the respective catchment area of the start point before it is deflected toward the corresponding stable endpoint in one of the near corners (e.g., trajectory c). In general, the sphere does not move along the shortest possible path to an endpoint. Only trajectories a and b on the mirror symmetry line are straight and can theoretically end at E_5_.

Note that the attractive force for the high-conductance sphere in front of the pointed electrode is almost twice as high as the repulsive force for the low-conductance sphere (trajectories a in Figure 8 and Figure 9). We suppose this is due to the superimposed attractive force of the mirror charges of the pointed electrode induced inside the high-conductance sphere.

#### 4.3.3. Relevance for DEP Measurements

The DEP mode was modeled to show how different drive modes may extend the application range of the four-pointed electrode chamber. The comparison shows that the conductance field and, thus the force field in DEP mode in the relevant area of the chamber is quite similar to that of the classical DEP chambers with plane-versus-pointed electrodes [1]. In the four-pointed electrode setup, the additional top and bottom electrodes generate additional attractive and repulsive forces for the high and low-conductance spheres, which are not observed in the plane-versus-pointed electrode chamber. In the case of the high-conductance sphere, the additional electrodes do introduce two additional end points with their own catchment areas. Appendix A presents an overlay of Figure 8 and Figure 9 showing the partly reversible DEP force in the chamber. It is obvious that the two additional electrodes help to shape the field in the central region of the chamber. Most likely, the gradient can be further improved by adjusting their drive voltage. However, such investigations are beyond the scope of this work. Even in the coarse (+++−) drive mode, the conductance fields along the line of symmetry are very similar to the plane-versus-pointed electrode configuration for both conductance cases (Figure 10).

When comparing the conductance fields for the high and low-conductance spheres in DEP mode, it is noticeable that both fields have an unstable saddle point on the symmetry line near the left electrode. For the high-conductance sphere, this point is also found in the plane-versus-pointed electrode configuration [1]. The conductance field near the left wall of the four-pointed electrode chamber is quite different from that in the plane-versus-pointed electrode chamber, where the plane counter electrode forms the left chamber wall. However, DEP measurements in front of the left chamber wall are not meaningful for either chamber. While the DEP forces near the left-pointed electrode are similar to those near the right-pointed electrode (Figure 10B), the DEP forces in front of the plane electrode are superimposed by mirror charge interactions [2].

In the plane-versus-pointed electrode chamber, DEP measurements would preferably be conducted along the symmetry line in the “dipole range” [1], which largely corresponds to the “reversibility range” in Figure 10C. These ranges feature force reversal for the conducting and nonconducting spheres and are roughly identical in the two chamber designs, i.e., in this region, the DEP mode mimics the classic DEP configuration with plane-versus-pointed electrodes. However, the degree of inhomogeneity of the object polarization depends on the size and position of the object, which affects the detection of internal structures, especially when the frequency dependence of the DEP forces is measured on relatively large objects (compare positions H, G, and C in Figure 6 and Figure 7, respectively).

The changing inhomogeneity affects the magnitude of the positive and negative DEP forces. It is therefore recommended to register positive and negative DEP velocities over the same distances within the reversibility range and to use the same chamber positions when using compensation methods, e.g., to detect the critical frequencies of DEP [12], which were later also referred to as “cross-over frequencies” [13].

### 4.4. (+−+−)—Drive Mode (Field-Cage Mode)

#### 4.4.1. Field Distribution and Chamber Conductance

In the “field-cage” or “trapping” mode, two electrodes at opposite edges are driven against the two electrodes at the other two opposite edges. Note that only low-conductance (low polarizable) objects but no high-conductance (high polarizable) objects are trapped at locations distant from the electrodes, i.e., at the center of the chamber.

Figure 11 shows the results for the 1.0-S sphere in the 0.1-S medium. However, the field-cage mode has simultaneous horizontal, vertical and diagonal mirror symmetry lines, increasing the number of sibling positions. While the center position G is unique, positions A through F each have three siblings. In Figure 11, the sphere is clearly inhomogeneously polarized at all positions. The conductance of the chamber increases in the order w/o < G < F < A < C < B < E < D. Accordingly, D and G are the most and least favorable positions, according to LMEP. The conductances of the seven positions are elements of the 160 × 160 conductance matrix (Appendix A). When equidistant current lines were chosen at x = 40 µm for calculating the equipotential and current lines, all current lines from the left negative electrode ended on either the top or bottom electrode. Additional runs were performed for the right half of the chamber to complete the plots.

Figure 12 shows the results for the 0.1-S sphere in a 1.0-S medium. The sphere positions were chosen as in Figure 11 because the symmetry and sibling properties were the same. Also, in Figure 12, the sphere is inhomogeneously polarized at all positions.

The conductance of the chamber increases in the order D < E < B < C < A < F < G < w/o. Accordingly, G and D are the most and least favorable positions, according to LMEP. The conductances of the seven positions are elements of the 160 × 160 conductance matrix (Appendix A).

#### 4.4.2. Trajectories and Forces

Figure 13 and Figure 14 show the conductance fields with the trajectories for the high and low-conductance spheres, respectively. The symmetry properties of the plots are as described above. Obviously, the field-cage mode has the highest number of mirror and rotational symmetries. The diagonals of the chamber from the lower left to upper right and from the upper left to lower right, and the horizontal and vertical lines are mirror planes or inverted mirror planes for the conductances, trajectories, and forces (Figure 1C).

For the high-conductance sphere, the two diagonals are watersheds that divide the chamber into four triangular catchment areas with the four stable endpoints E_2_, E_3_, E_4_ and E_5_ at the electrodes. Both diagonals have the common unstable endpoint E_1_ at their intersection in the chamber’s center. On both diagonals, there are two additional unstable saddle points near the intersections of the diagonals, with the lines connecting adjacent electrode tips (E_7_, E_9_ and E_6_, E_8_). From E_1_, eight straight trajectories run either to one of the four endpoints at the electrodes or along the diagonals to the unstable saddle points E_6_, E_7_, E_8,_ or E_9_. Trajectories along one of the four mirror symmetry lines have three siblings, e.g., b, n, j. Trajectories starting in the chamber volume (apart from mirror symmetry lines) are curved and have seven siblings. They all hit the chamber wall before running to the endpoints at the nearest electrode., e.g., a, c, d, e, f, g, h, i, k, l, m. Approaching the endpoint E_6_, the sphere moving along trajectories j or n may be deflected perpendicularly, depending on the nature of a possible slight perturbation, and then continue along either trajectory a or l, hitting the chamber wall near an electrode and generating a small force peak (Figure 13C).

The highest force peaks are observed at trajectories b and d, which hit the electrode directly and almost directly, respectively. The DEP force is zero at the unstable endpoint E1 and at the unstable saddle points E_6_, E_7_, E_8_, E_9_, but not at the stable endpoints E_2_, E_3_, E_4_, E_5_. Again, except for the direct electrode hit (trajectory b), from Equation (2) only minimal forces result for the end steps within the conductance field. Experimentally, when the sphere’s motion stops, the DEP force at the terminal point is compensated for by the pressure on the electrode.

The conductance field for the low-conductance sphere in Figure 14A has similar symmetry properties to that of the high-conductance sphere (Figure 13A), but with the directions of the forces and trajectories reversed. Accordingly, force peaks are observed when the sphere detaches from the chamber walls (trajectories a, e, and d). Forces are higher when the starting point of a trajectory is near an electrode (trajectories f and g) and highest when trajectories originate directly from an electrode (trajectories c and d).

The large central catchment area with central stable endpoint E_1_ is surrounded by four separate, nearly triangular catchment areas with stable endpoints E_2_–E_5_ at the corners of the chamber. Nearly straight watersheds separate the five catchment areas. The four watersheds have unstable saddle points (E_6_–E_9_) located at the intersection between the watersheds and the diagonals of the chamber. The effective DEP forces are zero at E_1_ and at the unstable saddle points E_6_–E_9_, but not at the stable endpoints E_2_–E_5_. As with the high-conductance sphere, trajectories along one of the four mirror symmetry lines have three siblings, for example, c, d, h, and i. In-plane trajectories have seven siblings, e.g., a, b, e, f, g. Within a radius of about 250 µm, the sphere is deflected straight or almost straight to E_1_, the central endpoint of the field cage.

Unlike in DEP mode, at the electrode, the repulsive forces for the low-conductance sphere are much larger than the attractive forces for the high-conductance sphere. However, this property is clearly advantageous for trapping experiments.

## 5. General Discussion

### 5.1. Conductance Fields, Electric Work and Dissipation

Modeling of AC-electrokinetic effects such as electroorientation, DEP, electrorotation, or mutual attraction is usually based on quasi-electrostatic approaches from the object’s perspective [12,13,14,15,16]. The approaches use lossy media properties, although they assume that the systems are in an equilibrium state without energy dissipation by resistive and displacement currents. Moreover, any electrokinetic movement must in itself lead to a dissipation of energy, and, surprisingly, the experimental observations and the models are in good agreement.

In general, objects that are higher and lower polarizable than the suspension medium are assumed to move in (positive DEP) or against (negative DEP) the direction of the field gradient. This view is largely correct, especially for small objects with a largely homogeneous polarization. They can “sense” and track the electric field gradient very locally and with little distortion of the external field. However, the description of their polarization with the Clausius-Mossotti factor, which assumes homogeneous object polarization, may become problematic in microchambers [17]. Inhomogeneous object polarization becomes more likely when the objects are relatively large with respect to the chamber [15,16,18,19,20,21]. In such cases, the total DEP force must be derived from the superposition of the polarization contributions of the entire object volume with the inhomogeneous external field.

We have shown that positive and negative DEP synchronously increases the total effective permittivity and conductivity, i.e., the total polarizability and dissipation of the DEP system [9]. Accordingly, the DEP force can be derived from the increase in dissipation of the electric field energy in ohmic heat or the capacitive charge work in the system. While a small portion of the energy causes the DEP translation, the translation increases the conductance of the DEP system. In turn, total electrical work and energy dissipation increase as the DEP progresses.

To model the position dependence of the dissipation, we have introduced the conductance field, which is the DC or low-frequency equivalent of the capacitance (work) field. In both fields, the energy field gradient describes the DEP forces and DEP trajectories.

For frequency-dependent models, especially for biological cells, the DEP system’s apparent (or complex) specific permittivity or conductivity can be described by the Maxwell-Wagner mixing equation [22,23]. After separating the reactive and active components of the capacitive charge work and dissipated energy, respectively, it was shown that the DEP is driven exclusively by the active components of the object’s polarization [9]. As in electrical machines, the reactive part is out-of-phase with the active component and performs no DEP work. The assumption of DC properties for the object and the external medium prevents problems in separating reactive contributions in the electric work conducted on the DEP system. However, this simplification does not reduce the complexity of the field-induced object behavior.

### 5.2. Thermodynamic Aspects

Previous work has compared the thermodynamic aspects of our new approach with existing approaches [8,9]. Since both the electric field and the DEP force are vector parameters, the LMEP approach does not violate the Curie-Prigogine principle, which prohibits scalar and vector quantities coupling in isotropic systems. Assuming thermal equilibrium under the condition of constant field strength, which is generally applied to DEP chambers, the entropy production corresponds to the dissipation of electrical energy, which is itself proportional to the conductivity or conductance of the DEP system and the square of the field strength [10].

One alternative assumption would be that the DEP system is close to equilibrium in a linear range and applying the electrode voltage causes only a slight deviation from equilibrium. Then, according to the Prigogine principle, DEP should cause the system to approach a new “field-on equilibrium” by minimizing entropy production [24,25,26]. However, the system is clearly nonlinear since its conductance and energy dissipation change with the field-induced DEP, even if thermal effects on parameters such as viscosities, conductivities, and permittivities are neglected. However, after the field is turned off, the Prigogine principle could regulate the path of the system back to the equilibrium state without entropy production and with a random position of the sphere. Nevertheless, the required diffusion of cell-size objects will take significantly longer than DEP.

In experiments and theory, DEP increases the overall polarizability and, more generally, decreases the system’s impedance. The corresponding increase in dissipation of electrical energy appears to be consistent with the Prigogine-Glansdorff principle, which allows only positive changes in entropy production due to an induced effect in dissipative structures and indicates that the system tends to follow the LMEP, i.e., the proposed fourth law of thermodynamics, which is contrary to the Prigogine principle [4,5,6,7].

Although our work shows that the LMEP provides a phenomenological criterion for AC-electrokinetic effects, entropy production, like many thermodynamic parameters, cannot directly explain the origin of the force at the level of individual objects [14].

## 6. Conclusions

Possibly, the main conclusion is that DEP itself can generally be viewed as a conditional polarization mechanism, even if it is slow in terms of the field oscillation in AC fields [8,9]. This argument is based on the more general view that positive and negative DEP are interpreted from the perspective of the object as displacement of a lower polarizable medium by a higher polarizable medium, where the higher polarizable medium can be either the object itself (positive DEP) or the external medium (negative DEP). From this consideration, it is immediately clear that the DEP for a suspension of objects must lead to an increase in polarizability for the whole system. This is the basis for modeling DEP in a given system by combining the system’s approach with the “conductance field”, the “capacitance field” or more generally, a “polarizability field” of the system. The fields describe the conductance of the system as a function of the object’s position in the DEP chamber. In the system, the DEP translation of objects follows field gradients while being retarded by the viscous properties of the suspension.

Calculations with smaller and larger spheres in the DEP systems have shown that the properties of the chambers discussed here as watersheds, saddle points, and catchment areas are qualitatively identical over a wide range of relative sizes of chamber to sphere. This was also found in initial calculations on 3D models, which require considerably more computing time than the 2D models.

The system’s perspective of DEP allows the consideration of an inhomogeneous object polarization and its interaction with an inhomogeneous external field. The calculation of the resulting interaction forces is simplified since the forces can directly be derived from differences in the system’s conductance (or capacitance). This permits a description of the contributions of effects such as induced multipoles or inhomogeneities of the external field induced only by the object’s presence, etc., which are tedious to model in object approaches. Moreover, the attractive forces between neighboring objects can easily be modeled and compared with analytical multipole models in which each object is subjected to the field created by the inhomogeneous polarization of the other object [27].

Up to now, it has been shown that the LMEP approach works for the electroorientation of ellipsoidal objects [8], for DEP in multiple electrode configurations and for the calculation of mirror charge-induced forces [1,2,9]. For example, LMEP modeling of AC-electrokinetic effects can be extended to nonspherical objects, multibody systems, or Janus particles [28] to compute combined orientation, translation, and aggregation patterns [14,16,20]. Other examples include objects inducing attractive mirror charges on flat electrode surfaces and pointed electrodes inducing mirror charges in large objects. Another problem is the induction of repulsive mirror charges on nonconducting chamber walls [2]. However, the superposition of DEP translation with induced fluid currents can complicate the registration of object trajectories in real systems.

The system approach represents a significant simplification for numerical modeling of DEP force fields in microfluidic systems. This can be helpful in the search for optimal electrode drive modes for the manipulation and positioning of biological objects and colloidal particles [13,14,15,17,18,19,20,21,23]. The four-electrode system considered here can be readily used for electro-rotation of objects [11,13,15,21]. The selective switching of drive voltages and AC phases at the electrodes can be used to manipulate and position the object in multi-electrode chambers [29,30].

In a subsequent paper, we plan to investigate the experimental behavior of cells and colloidal objects in microchambers with four electrodes and compare it with the behavior expected from to our new theory. However, especially for negative DEP, it has already been observed that the DEP forces in some areas, such as near saddle points, are minimal and may not overcome sedimentation, surface friction, and subsequent adhesion.

## Figures and Tables

**Figure 1 micromachines-14-02042-f001:**
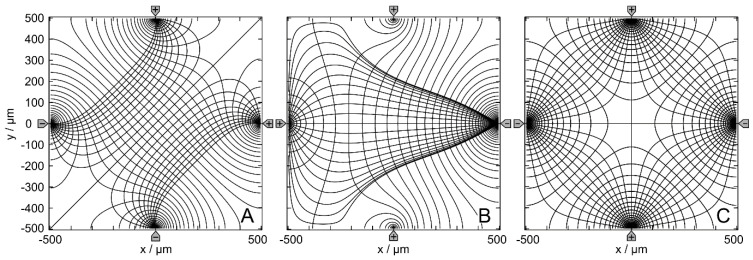
Potential and current line distributions in the 1 × 1-mm chamber without the object. The four-pointed electrodes were energized with 0.5 V (marked by “+”; in AC-drive, this corresponds to the 180°-phase) versus −0.5 V at the counter electrodes (marked by “−“; in AC-drive, this corresponds to the −180°-phase). For a clearer presentation of the current and potential distributions, equidistant current lines were used at x = 0 in A and x = −250 µm in B. For C, two distributions with equidistant current lines at x = −250 µm and at x = 250 µm were combined. The calculated basic sheet conductances $L_{Basic}^{2D}$ for the 100 mS//1 S media are (**A**): 42.31 mS//422.9 mS, (**B**): 32.75 mS//327.4 mS and (**C**): 46.67 mS//466.5 mS, corresponding to cell constants $k^{2D}$ of approx. 0.423, 0.327 and 0.466, respectively.

**Figure 2 micromachines-14-02042-f002:**
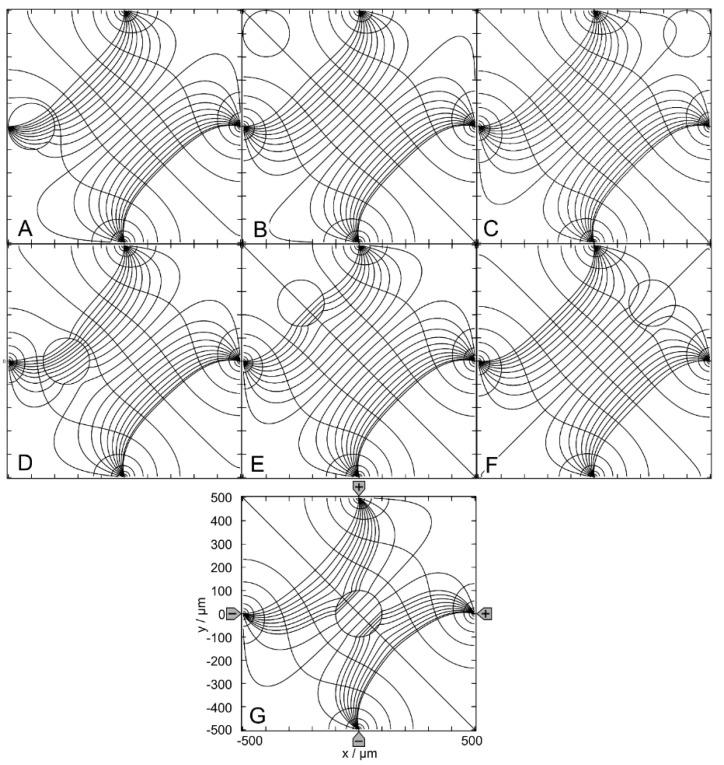
Potential and current line distributions for different positions of the 1.0-S sphere in 0.1-S medium for the (++−−)—drive mode. The position of the electrodes is sketched in G only. The overall conductances of the chamber are (**A**): 51.62 mS, (**B**): 42.50 mS, (**C**): 42.31 mS, (**D**): 43.09 mS, (**E**): 43.07 mS, (**F**): 42.46 mS and (**G**): 42.94 mS (Table 1). The current lines were chosen to be equidistant at x = 0.

**Figure 3 micromachines-14-02042-f003:**
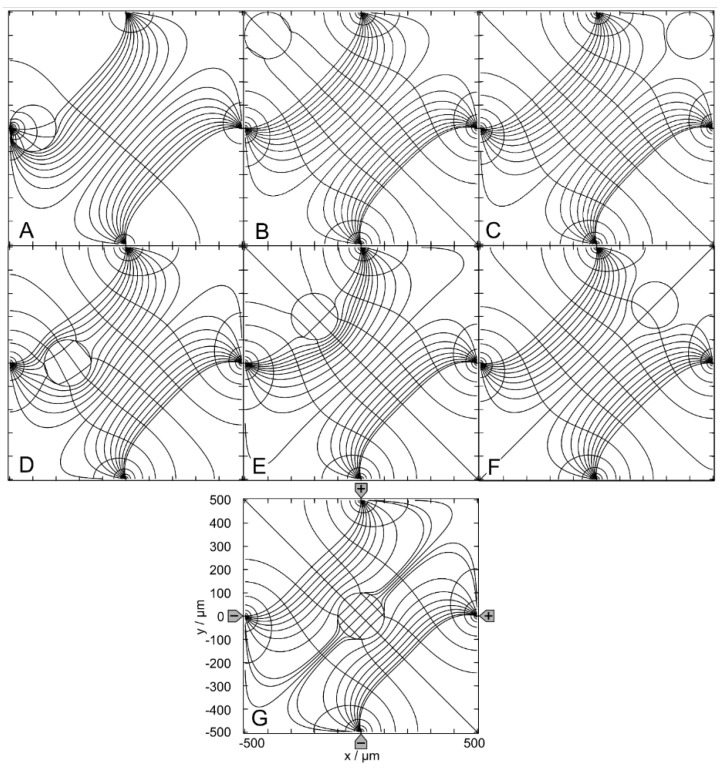
Potential and current line distributions for different positions of the 0.1-S sphere in 1.0-S medium for the (++−−)—drive mode. The position of the electrodes is sketched in G only. The conductances of the chamber are (**A**): 173.0 mS, (**B**): 420.1 mS, (**C**): 422.9 mS, (**D**): 414.6 mS, (**E**): 414.3 mS, (**F**): 421.2 mS and (**G**): 416.4 mS (Table 1). The current lines were chosen to be equidistant at x = 0.

**Figure 4 micromachines-14-02042-f004:**
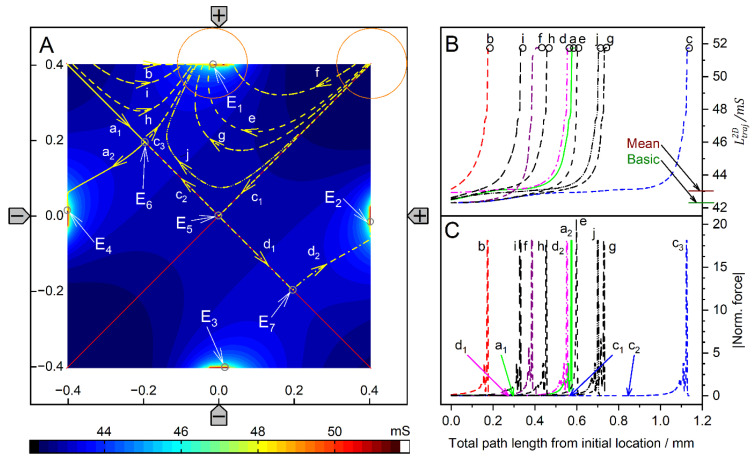
Single 200-µm, 1.0-S sphere (reddish circles in (**A**)) in the chamber of Figure 1A with 0.1-S medium. (**A**): Conductance field plot with trajectories (a–j). Two watersheds (two diagonal red lines) separate the four catchment areas of the stable endpoints E_1_, E_2_, E_3_ and E_4_. E_5_, E_6_ and E_7_ are unstable saddle points on one of the watersheds. (**B**): Chamber conductance along the trajectories. The basic, minimum, mean, and maximum conductances are 42.31 mS (w/o sphere), 42.31 mS (Figure 2C), 42.03 mS, and 51.75 mS (E_1_–E_4_, Figure 2A), respectively (Table 1). Trajectories c_1_, c_2_, and d_1_ run on watersheds and through unstable endpoints. (**C**): Normalized DEP forces calculated from the conductance values in (**B**).

**Figure 5 micromachines-14-02042-f005:**
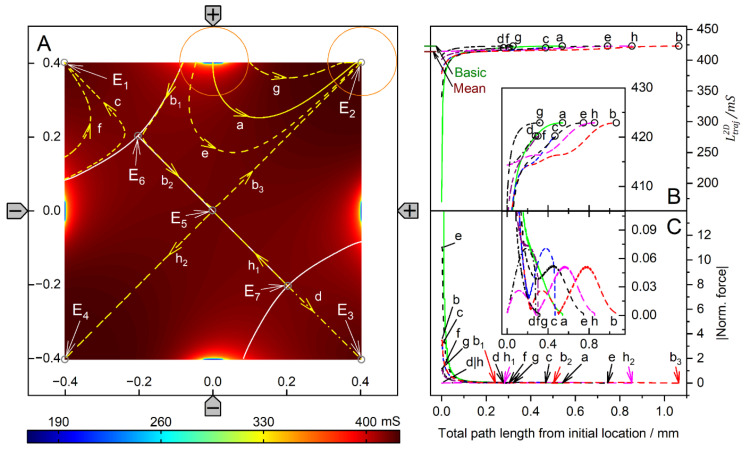
Single 200-µm 2D sphere of 0.1-S (reddish circles in (**A**)) in the chamber of Figure 1 with 1.0-S medium. (**A**): Conductance field plot with trajectories (a–h). Three watersheds (the diagonal white line between E_6_ and E_7_ and two curved lines through E_6_ and E_7_ at the end of the diagonal watershed) separate the four catchment areas of the stable endpoints E_1_, E_2_, E_3_ and E_4_. E_5_, E_6,_ and E_7_ are unstable saddle points. (**B**): Chamber conductance along the trajectories. The basic, minimum, mean, and maximum conductances are 422.9 mS (w/o sphere), 169.7 mS, 414.1 mS, and 422.9 mS (E_1_, E_2_, E_3_ and E_4_, Figure 3C), respectively (Table 1). Trajectories b_1_, b_2_, and h_1_ follow watersheds and run across unstable endpoints. (**C**): Normalized DEP forces calculated from the conductance values in (**B**). The initial force peak of 75.35 for trajectory a (green) was truncated, and the ordinate was shortened to increase the resolution for all other trajectories.

**Figure 6 micromachines-14-02042-f006:**
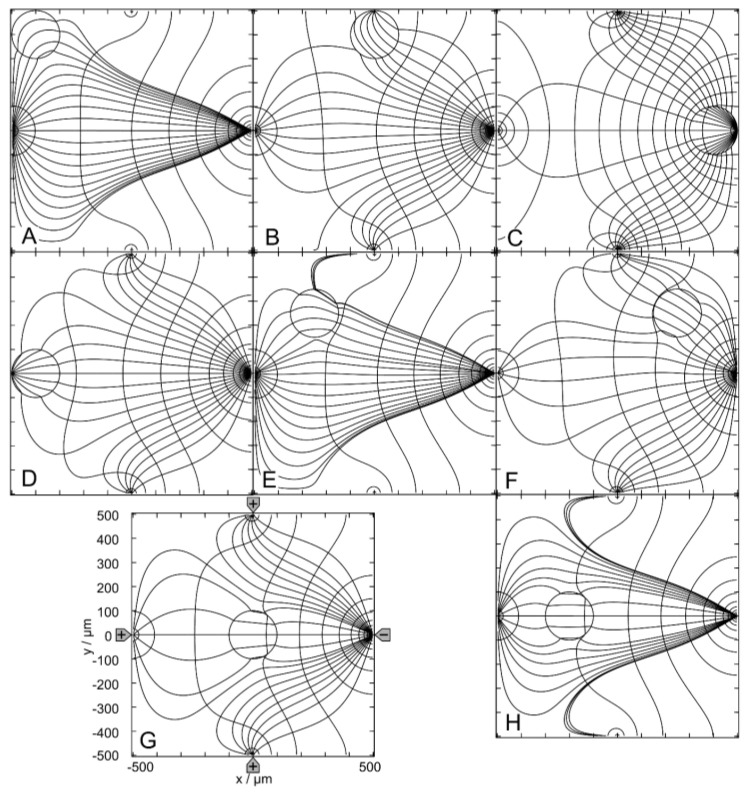
Potential and current line distributions for different positions of the 1.0-S sphere in 0.1-S medium for the (+++−)—drive mode. The position of the electrodes is sketched in (**G**) only. The conductances of the chamber are (**A**): 32.76 mS, (**B**): 35.03 mS, (**C**): 72.19 mS, (**D**): 34.90 mS, (**E**): 32.82 mS, (**F**): 33.34 mS, (**G**): 33.09 mS and (**H**): 32.96 mS (Table 1). The current lines were chosen to be equidistant at x = −300 µm (**A**), x = −250 µm (**E**), x = −200 µm (**H**), x = 50 µm (**B**,**D**,**G**), x = 250 µm (**F**) and x = 300 µm (**C**).

**Figure 7 micromachines-14-02042-f007:**
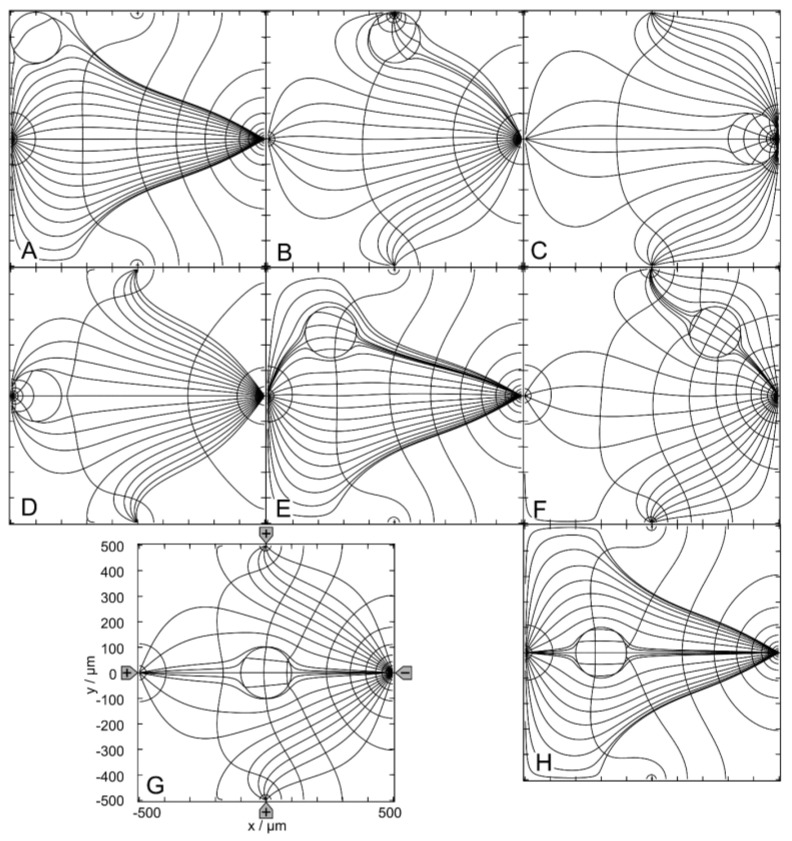
Potential and current line distributions for different positions of the 0.1-S sphere in 1.0-S medium for the (+++−)—drive mode. The position of the electrodes is sketched only in (**G**). The conductances of the chamber are (**A**): 327.3 mS, (**B**): 217.9 mS, (**C**): 60.02 mS, (**D**): 218.8 mS, (**E**): 326.6 mS, (**F**): 321.0 mS, (**G**): 323.8 mS and (**H**): 325.2 mS (Table 1). The current lines were chosen to be equidistant at x = −300 µm (**A**), x = −250 µm (**E**), x = −200 µm (**H**), x = 50 µm (**B**,**D**,**G**), x = 250 µm (**F**) and x = 300 µm (**C**).

**Figure 8 micromachines-14-02042-f008:**
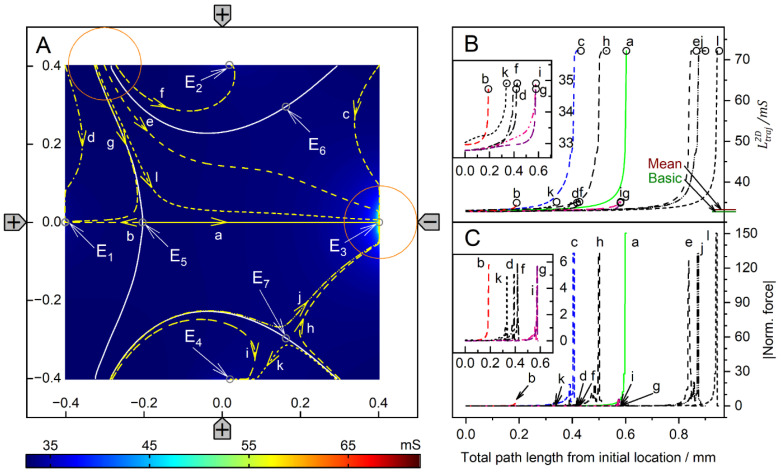
Single 200-µm, 1.0-S sphere (reddish circles in (**A**)) in the chamber of Figure 1B with 0.1-S medium. (**A**): Conductance field plot with trajectories (a–k). Three watersheds (curved white lines) with unstable saddle points E_5_, E_6_, and E_7_ separate four catchment areas for the stable endpoints E_1_, E_2_, E_3_, and E_4_. (**B**): Chamber conductance along the trajectories. The basic, minimum, mean, and maximum conductances are 32.75 mS (w/o sphere), 32.76 mS (Figure 6A), 33.27 mS, and 72.19 mS (E_3_; Figure 6C), respectively (Table 1). (**C**): Normalized DEP forces calculated from the conductance values in (**B**). The arrows mark the ends of trajectories.

**Figure 9 micromachines-14-02042-f009:**
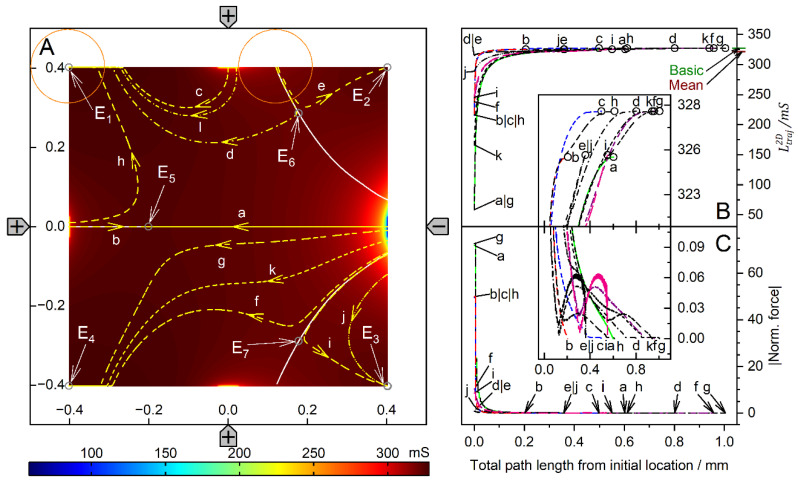
Single 200-µm 2D sphere of 0.1-S (reddish circles in (**A**)) in the chamber of Figure 1B with 1.0-S medium. (**A**): Conductance field plot with trajectories (a–k). Three watersheds (horizontal symmetry line, two curved white lines) with the unstable saddle points E_5_, E_6_, and E_7_ separate four catchment areas for the stable endpoints E_1_, E_2_, E_3_, and E_4_. (**B**): Chamber conductance along the trajectories. The basic, minimum, mean, and maximum conductances are 327.4 mS (w/o sphere), 58.45 mS (Figure 7C), 321.4 mS, and 327.3 mS (E_1_, E_4_; Figure 7A), respectively (Table 1). (**C**): Normalized DEP forces calculated from the conductance values in (**B**). The arrows mark the ends of trajectories.

**Figure 10 micromachines-14-02042-f010:**
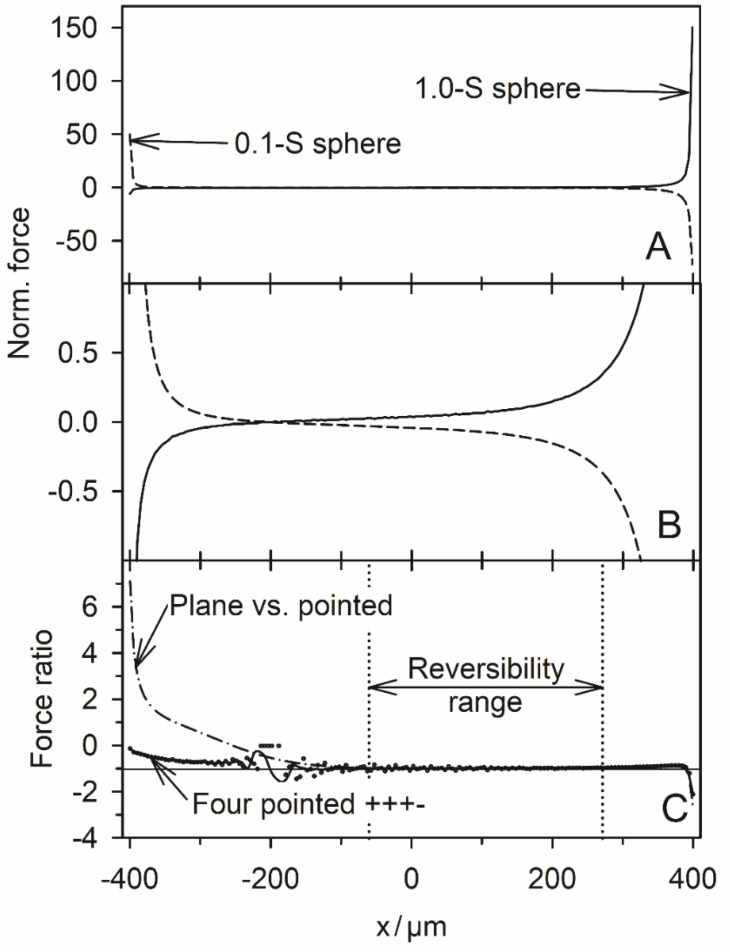
Normalized DEP forces and DEP force reversibility along the chamber’s symmetry line in the DEP mode. (**A**): Normalized DEP forces acting on the 1.0-S (full line, corresponding to trajectories a and b in Figure 8) and 0.1-S spheres (dashed line, corresponding to trajectories a and b in Figure 9). (**B**): Zoom of A. The forces vanish for x = −201 µm for both spheres at the bifurcation points E_5_ (Figure 8 and Figure 9). (**C**): DEP force reversibility as calculated from the quotient of the normalized DEP forces on the 1.0-S and 0.1-S spheres using the data of Appendix A, which are summarized in Appendix A). The horizontal line marks the force ratio of −1, i.e., ideal reversibility, which can be assumed to be within the “reversibility range” −60 µm < x < 270 µm.

**Figure 11 micromachines-14-02042-f011:**
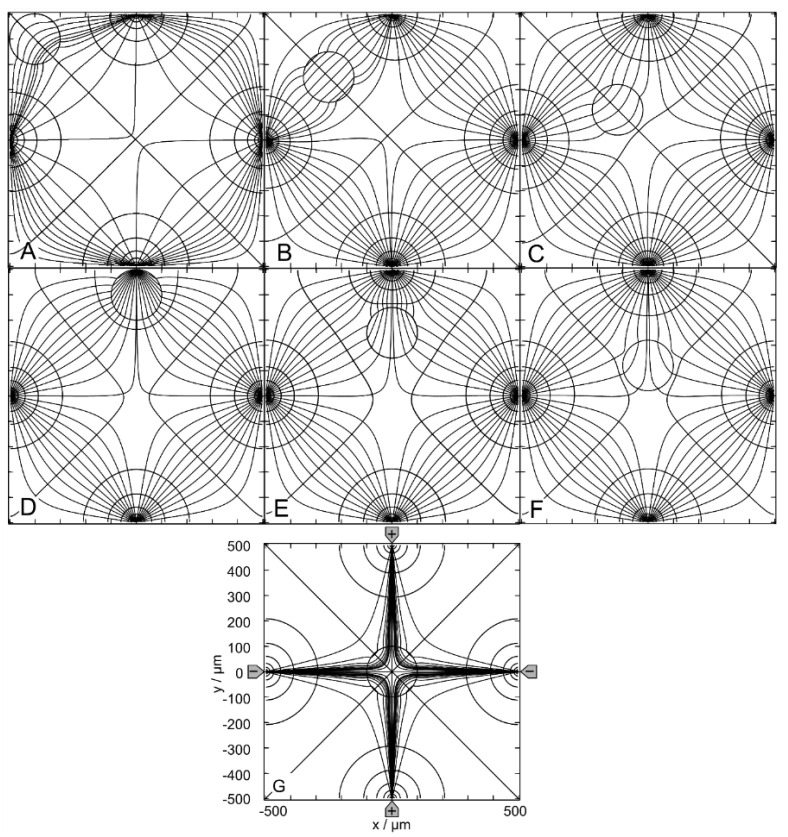
Potential and current line distributions for different positions of the 1.0-S sphere in 0.1-S medium. The location of the electrodes is sketched only in (**G**). The conductances of the chamber are (**A**): 46.78 mS, (**B**): 47.04 mS, (**C**): 46.84 mS, (**D**): 57.85 mS, (**E**): 47.10 mS, (**F**): 46.77 mS and (**G**): 46.69 mS (Table 1). Equipotential and current lines were combined from separate calculations for the left and right halves of the chamber, with current lines chosen to be equidistant at x = −40 µm and x = 40 µm, respectively.

**Figure 12 micromachines-14-02042-f012:**
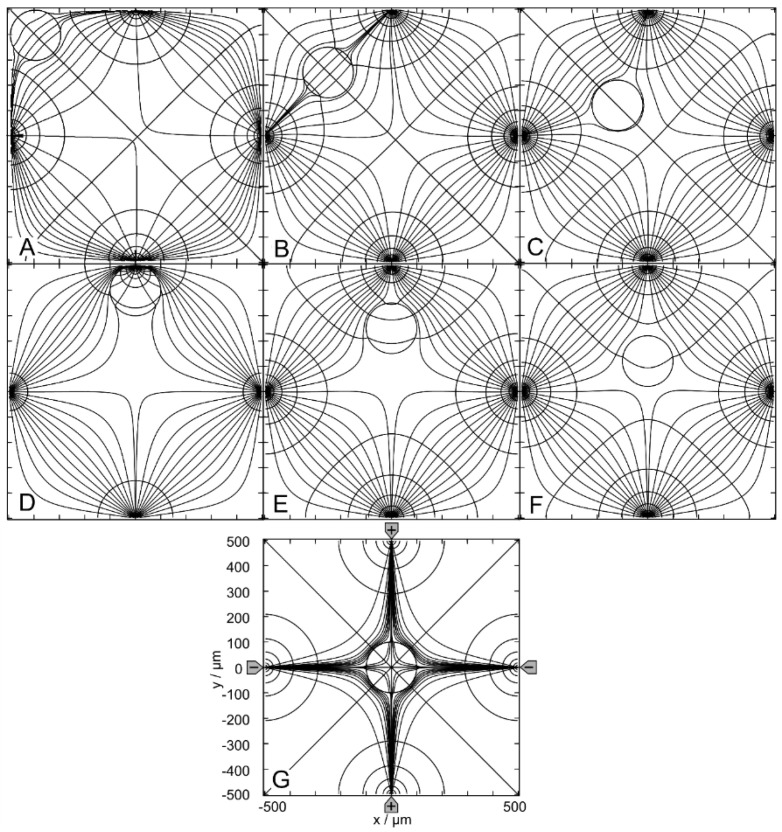
Potential and current line distributions for different positions of the 0.1-S sphere in 1.0-S medium. The position of the electrodes is sketched only in (**G**). The conductances of the chamber are (**A**): 464.8 mS, (**B**): 462.3 mS, (**C**): 464.7 mS, (**D**): 180.5 mS, (**E**): 461.8 mS, (**F**): 465.4 mS and (**G**): 466.2 mS (Table 1). Equipotential and current lines were combined from separate calculations for the left and right halves of the chamber, with current lines chosen to be equidistant at x = −40 µm and x = 40 µm, respectively.

**Figure 13 micromachines-14-02042-f013:**
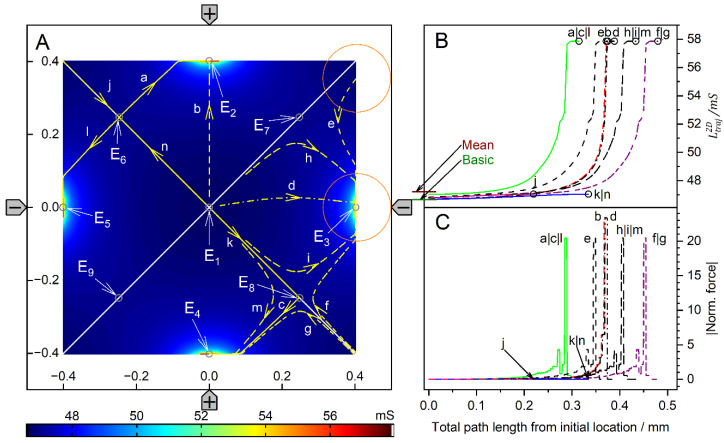
Single 200-µm, 1.0-S sphere (reddish circles in (**A**)) in the chamber of Figure 1C with 0.1-S medium. (**A**): Conductance field plot with trajectories (a–n). Two diagonal watersheds separate the four catchment areas of the stable endpoints E_2_, E_3_, E_4_ and E_5_. E_1_ is a single unstable point precisely in the chamber’s center. (**B**): Chamber conductance along the trajectories. The basic, minimum, mean, and maximum conductances are 46.67 mS (w/o sphere), 46.69 mS (Figure 11G), 47.21, and 57.85 mS (E_2_, E_3_, E_4_, E_5_; Figure 11D). (**C**): Normalized DEP forces calculated from the conductance values in (**B**). The arrows mark the end of the trajectories.

**Figure 14 micromachines-14-02042-f014:**
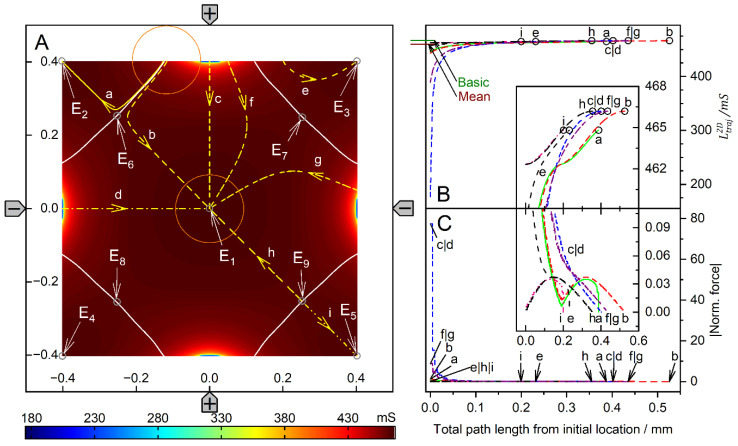
Single 200-µm 2D sphere of 0.1-S (reddish circles in (**A**)) in the chamber of Figure 1C with 1.0-S medium. (**A**): Conductance field plot with trajectories (a–i). Four watersheds (white lines cutting off the chamber’s corners) separate the five catchment areas of the stable endpoints E_1_–E_5_. E_6_–E_9_ are unstable saddle points in the middle of the watersheds. (**B**): Chamber conductance along the trajectories. The basic (w/o sphere), minimum, mean, and maximum conductances are 466.5 mS, 176.9 mS (Figure 12D), 459.3 mS, and 466.2 mS (E_1_; Figure 12G), respectively. (**C**): Normalized DEP forces calculated from the conductance values in (**B**).

**Table 1 micromachines-14-02042-t001:** Conductances of the chambers for the drive modes (++−−), (+++−), and (+−+−) (Figure 1) without (w/o sphere) and with the 2D sphere at different positions in the chamber (A–H) in mS. The conductances of the spheres and the outside chamber combine either 1.0 S for the sphere with 0.1 S for the medium (1.0 in 0.1) or 0.1 S for the sphere with 1.0 S for the medium (0.1 in 1.0).

Mode	++−−		+++−		+−+−	
Conductance	1.0-in-0.1	0.1-in-1.0	1.0-in-0.1	0.1-in-1.0	1.0-in-0.1	0.1-in-1.0
Basic (w/o sphere)	42.3103	422.9419	32.7518	327.3895	46.6665	466.4686
Minimum	42.3132	169.6693	32.7567	58.4465	46.6926	176.8777
Mean	43.0311	414.1074	33.2723	321.3707	47.2079	459.3097
Maximum	51.7546	422.9101	72.1644	327.3172	57.8512	466.1858
A	51.6242	172.9551	32.7572	327.3137	46.7797	464.7503
B	42.4985	420.1077	35.0269	217.9206	47.0445	462.2856
C	42.3139	422.9059	72.1854	60.0215	46.8354	464.6568
D	43.0865	414.6169	34.9014	218.8037	57.8507	180.4669
E	43.0702	414.2677	32.8186	326.6342	47.0987	461.7775
F	42.4621	421.2087	33.3379	321.0329	46.7698	465.3648
G	42.9407	416.3622	33.0919	323.8072	46.6926	466.1859
H	--	--	32.9595	325.1618	--	--

## Data Availability

Not applicable.

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
