# Peer review of "Trajectories and Forces in Four-Electrode Chambers Operated in Object-Shift, Dielectrophoresis and Field-Cage Modes—Considerations from the System’s Point of View"

_micromachines, 2023, doi:10.3390/mi14112042_

Round 1
Reviewer 1 Report
Comments and Suggestions for Authors
The paper treats theoretically, in a novel way, the dielectrophoretic (DEP) behavior of spherical microparticles of different conductivities located within a four-electrode chamber operating in 3 different modes, including DEP-, object-shift- and trap modes. Unlike earlier approaches (commonly based on the local DEP-force considerations), the new systemic approach applied in this study is based on the law of maximum entropy production. Using this approach, which accounts for inhomogeneous field distribution in both the chamber and the particle, the authors were able to calculate very accurately the DEP forces and the corresponding particle trajectories.
The paper is well written. The analysis and discussion of the theoretical data are pertinent and convincing. The numerical results, e.g., DEP forces, particles trajectories, chamber conductance, etc., are well illustrated by figures and tables.
Despite its theoretical nature, the paper offers several quantitative predictions which should stimulate further experimental work on the basic mechanisms of the interaction of electromagnetic fields with biological cell and colloidal particles. The paper is also of great interest to researchers working in the fields of membrane and cell biophysics, using AC electrokinetic and related dielectric spectroscopic techniques.
Author Response
Dear Reviewer,
Thank you for your time, your encouraging comments and your very positive recommendation. We thoroughly went over the text in the “trace changes” mode and corrected a number of typos. We have highlighted the changes that were made in relation with an additional reference, which was introduced according to a recommendation by the second reviewer.
Sincerely,
Jan Gimsa
Reviewer 2 Report
Comments and Suggestions for Authors
This work is the third in a series of papers on the DEP behavior of high and low conductivity 2D spheres in square chambers driven by various combinations of idealized pointed and plane electrodes. In current study, the authors complete their considerations using configurations with four-pointed electrodes placed in the center of the square chamber’s edges. The four electrodes were operated in three drive modes; object-shift mode, DEP mode, and field-cage mode. The research topic is quite interesting and important. The manuscript is well written, and its structure is well organized. The scientific content is substantial, and discussions and anal yzes well support the conclusion. To this end, the referee here recommended its acceptance after a minor revision. Some minor points:
1. What is the frequency range in which the proposed method is applicable? Can the phenomenon of electrode polarization be considered?
2. A direct comparison of the DEP stress calculation between the currently proposed method and the full Maxwell stress tensor approach may be quite useful in terms of simulation validation.
3. One recent literature [Micromachines 2020, 11(3), 289] on numerical simulation of particle DEP motion in microfluidic channels may have some reference value for this paper.
Author Response
Dear Reviewer,
Thank you for your time, your encouraging comments and your very positive recommendation. We thoroughly went over the text in the “trace changes” mode and corrected a number of typos. We have highlighted the changes that were made also according to your remarks. Below are our responses to your points in the order in which you raised them.
1.1. What is the frequency range in which the proposed method is applicable?
Reply: There is no restriction in the frequency range. However, the application of the LMEP method is more simple when the object polarization consists only of active components and no out-of-phase (imaginary) components occur. This is the case for the low (conductivity) and high frequency (permittivity) limits. In the manuscript, this problem is mentioned under ”Theory”: “Unfortunately, in the overall system, in-phase contributions to energy dissipation can also arise from the interaction of the out-of-phase components of the currents with out-of-phase components of the induced polarizations. Moreover, these contributions to the total dissipation may depend on the object’s position. DEP force calculations from energy differences must exclude such contributions, which are not trivial [9].” In reference [9], the problem has been solved using a linear DEP model for a spherical object.
1.2. Can the phenomenon of electrode polarization be considered?
Reply: Yes, by introducing appropriate electrode properties. However, in the LMEP approach this would probably not be easier than in any other approach.
- A direct comparison of the DEP stress calculation between the currently proposed method and the full Maxwell stress tensor approach may be quite useful in terms of simulation validation.
Reply: You are right. However, when we compared our approach with the Maxwell stress tensor approach for a spherical model using COMSOL™, we found slight deviations. Unfortunately, we were not able to localize their source, yet.
- One recent literature [Micromachines2020, 11(3), 289] on numerical simulation of particle DEP motion in microfluidic channels may have some reference value for this paper.
Reply: As a response to you remark, we changed a paragraph in the conclusion section by introducing references for the different DEP phenomena. For Janus particles, we used the manuscript you proposed from the DEP series of Micromachines (now ref. 28).
Sincerely,
Jan Gimsa